# CNS Involvement of DLBCL Presenting with an Unusual Non-Enhancing Infiltrative Mass

**DOI:** 10.3390/diagnostics13223424

**Published:** 2023-11-10

**Authors:** Fu-Sheng Hsueh, Hung-Chieh Chen, Huey-En Tzeng

**Affiliations:** 1Department of Radiology, Taipei Veterans General Hospital, Taipei 11217, Taiwan; 2Department of Radiology, Division of Neuroradiology, Taichung Veterans General Hospital, Taichung 40705, Taiwan; 3School of Medicine, National Yang-Ming Chiao Tung University, Taipei 112304, Taiwan; 4Department of Internal Medicine, Division of Hematology and Oncology, Taipei Medical University Hospital, Taipei 110301, Taiwan; tzhuen@vghtc.gov.tw; 5Department of Medical Research, Taichung Veterans General Hospital, Taichung 40705, Taiwan; 6Department of Medicine, Division of Hematology/Medical Oncology, Taichung Veterans General Hospital, Taichung 40705, Taiwan; 7Ph.D. Program for Cancer Molecular Biology and Drug Discovery, Graduate Institute of Cancer Biology and Drug Discovery, College of Medical Science and Technology, Taipei Medical University, Taipei 110301, Taiwan; 8Department of Post-Baccalaureate Medicine, College of Medicine, National Chung Hsing University, Taichung 402202, Taiwan

**Keywords:** CNS lymphoma, MRI, CT

## Abstract

Central nervous system (CNS) involvement in diffuse large B-cell lymphoma (DLBCL) is relatively uncommon, occurring in approximately 5% of cases, with the majority of instances manifesting during relapse and often associated with poor prognoses. The aim of this case report is to present a unique occurrence of non-enhancing relapse of CNS lymphoma. Significantly, the patient had recently encountered a disease involvement in the axilla region, and subsequent to scheduled chemotherapy, she developed persistent neurological symptoms, leading to the discovery of a relapse of the CNS lymphoma. Our focus will be on delineating the clinical presentation, elucidating the findings observed in clinical imaging, and detailing the therapeutic approaches employed in this specific case. By highlighting these aspects, we aim to provide valuable insights into the diagnosis of the atypical presentation of CNS lymphoma.

A 61-year-old woman presented to our emergency department with a fever, reaching up to 38.5 degrees Celsius, accompanied by general malaise, poor appetite, and muscle soreness. She had a medical history of diffuse large B-cell lymphoma (DLBCL) which presented as multiple axillary lymph nodes, characterized by double expression and stage IIA. She had recently completed the fourth cycle of R-CHOP chemotherapy (rituximab, cyclophosphamide, doxorubicin, vincristine, prednisone) just nine days prior. Notably, she denied experiencing symptoms such as cough, sputum production, dyspnea, chest pain, abdominal pain, nausea, vomiting, diarrhea, tarry stools, dysuria, or joint pain. Additionally, no apparent skin rashes or wounds were observed during the examination.

Upon evaluation, her fever persisted at 38.2 degrees Celsius, and laboratory investigations revealed leukopenia with a white blood cell count (WBC) of 610/μL. The urinalysis showed no evidence of pyuria, and the chest X-ray results were unremarkable. The treatment included the administration of Granulocyte Colony-Stimulating Factor (GCSF) for leukopenia and empirical antibiotics, specifically piperacillin/tazobactam, to address the neutropenic fever. An extensive infectious workup was conducted on admission, including a cerebrospinal fluid (CSF) analysis, which did not identify a clear infectious focus. The blood cultures yielded no bacterial growth. The patient’s WBC count elevated to 7430/μL after GCSF treatment. However, intermittent fever was still observed. Six days after admission, she exhibited a slow response and developed right-sided weakness. A relapse of CNS lymphoma or other CNS lesion was suspected. A brain computed tomography (CT) scan revealed abnormal hypointense changes in the left frontal lobe, left basal ganglion, left thalamus, and left cerebral peduncle with associated brain swelling (Figure 1). The subsequent magnetic resonance imaging (MRI), performed three days later, revealed hyperintensity in the left frontal and left parietal lobes, bilateral corona radiata, basal ganglia, and thalami, as well as the left temporal lobe and left-side brainstem on T2 weighted images (T2WIs), fluid attenuated inversion recovery (FLAIR), and diffusion weighted images (DWIs) without obvious restricted diffusion. No definite contrast enhancement was observed on the post-gadolinium T1 weighted images (Gd-T1WIs) (Figure 2). The presence of a mass effect from the infiltrative lesions, resulting in the compression of the left lateral ventricle and a midline shift to the right side, was observed. Notably, magnetic resonance spectroscopy (MRS) focused on the left basal ganglion lesion showed an elevated choline peak and decreased N-acetylaspartate (NAA) peak, with the presence of a lactate peak (Figure 3). A stereotactic brain biopsy for the left basal ganglion lesion and the left frontal lobe lesion confirmed the presence of diffuse large B-cell lymphoma, and the patient subsequently received whole-brain radiotherapy. Following treatment, the patient’s consciousness improved to an alert state (E4V5M6), and her right limb muscle strength showed improvement. Under stable conditions, she underwent another cycle of R-CHOP chemotherapy and completed whole-brain radiotherapy (WBRT). The patient was transferred to a nursing home under a stable clinical condition.

Diffuse large B-cell lymphoma (DLBCL) is the most common lymphoid malignancy in adults globally, accounting for 40 percent of all non-Hodgkin lymphoma cases. The standard first-line treatment for DLBCL is the R-CHOP regimen, which has significantly improved cure rates. Nevertheless, about 40 percent of DLBCL patients experience relapse or refractory disease [1]. CNS relapses in DLBCL are rare but have devastating complications, occurring in 2–5 percent of overall cases [2]. In double expression lymphoma, the incidence of CNS relapse reaches 9.7 percent at 2 years [3].

Radiological findings in CNS lymphoma can vary significantly. In immunocompetent patients, the typical presentation involves single or multiple homogeneously enhanced lesions in deep brain structures [4,5]. Conversely, immunocompromised patients may exhibit irregular rim-enhancing lesions or infiltrative enhancing lesions [6]. Approximately 25% of patients may display atypical radiologic findings [7], which can complicate the diagnosis of lymphoma, mimicking other conditions such as glioblastoma [8] or even neuroinflammatory diseases [9]. Utilizing the characteristics of a low relative cerebral blood volume (rCBV) on perfusion-weighted images and low apparent diffusion coefficient (ADC) values can be helpful in differentiating CNSL from other disease entities [10]. Non-enhancing CNS lymphoma is particularly rare and has been limited to case reports [11]. The delayed diagnosis of CNS lymphoma is often a contributing factor to poor prognoses.

In this report, we present a unique case of CNS lymphoma characterized by an infiltrative lesion with poor enhancement during recurrence. Interestingly, most of the lesions in our case exhibited a high signal intensity on T2WI without reduced ADC values. Notably, we did not perform perfusion-weighted imaging, rendering this entire presentation atypical for CNS lymphoma relapse. Common differential diagnoses for infiltrative, non-enhancing brain lesions include low-grade glioma, encephalitis, or progressive multifocal leukoencephalopathy (PML). Extensive encephalitis involvement typically leads to impaired consciousness, while PML lesions in the brain usually do not cause mass effects [12]. These features are unlikely to be present in our case, making the diagnoses of encephalitis or PML improbable. Stereotactic brain biopsy was conducted because we could not differentiate between a low-grade glioma or an unusual presentation of CNS lymphoma.

In conclusion, we emphasize the significance of this case report in highlighting the importance of CNS lymphoma. While CNS lymphoma is a rare condition, especially when presenting with an infiltrative non-enhancing mass, it should be considered in patients with active DLBCL or even those at high risk of double expression lymphomas. We hope this article can serve as a valuable resource and contribute to a better understanding of this condition.

## Figures and Tables

**Figure 1 diagnostics-13-03424-f001:**
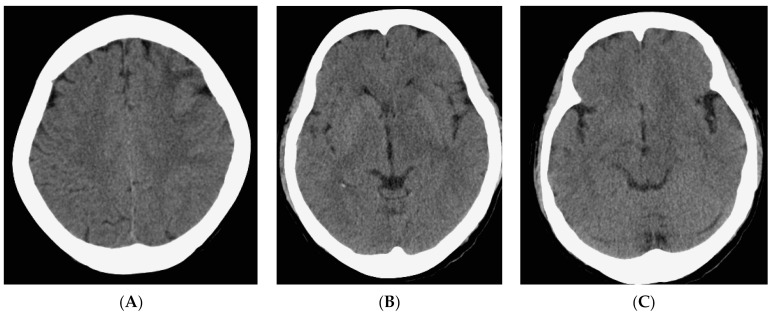
Non-contrast CT scan of brain. The images show a hypointense area with brain swelling at left frontal lobe (**A**), left basal ganglion (**B**), left thalamus (**B**) and left cerebral peduncle (**C**).

**Figure 2 diagnostics-13-03424-f002:**
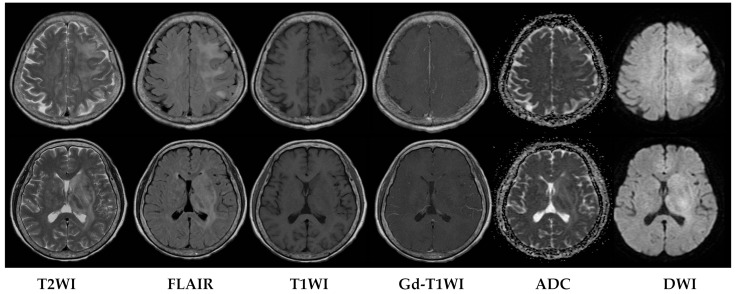
MRI performed three days after CT scan. MRI fused images of axial T2WI, FLAIR, T1WI, Gd-T1WI, ADC, and DWI images. Abnormal hyperintense lesions are seen at left frontal lobe, left parietal lobe, bilateral corona radiata, bilateral basal ganglia, bilateral thalami on FLAIR, DWI, and ADC images. No obvious contrast enhancement or restricted diffusion is seen.

**Figure 3 diagnostics-13-03424-f003:**
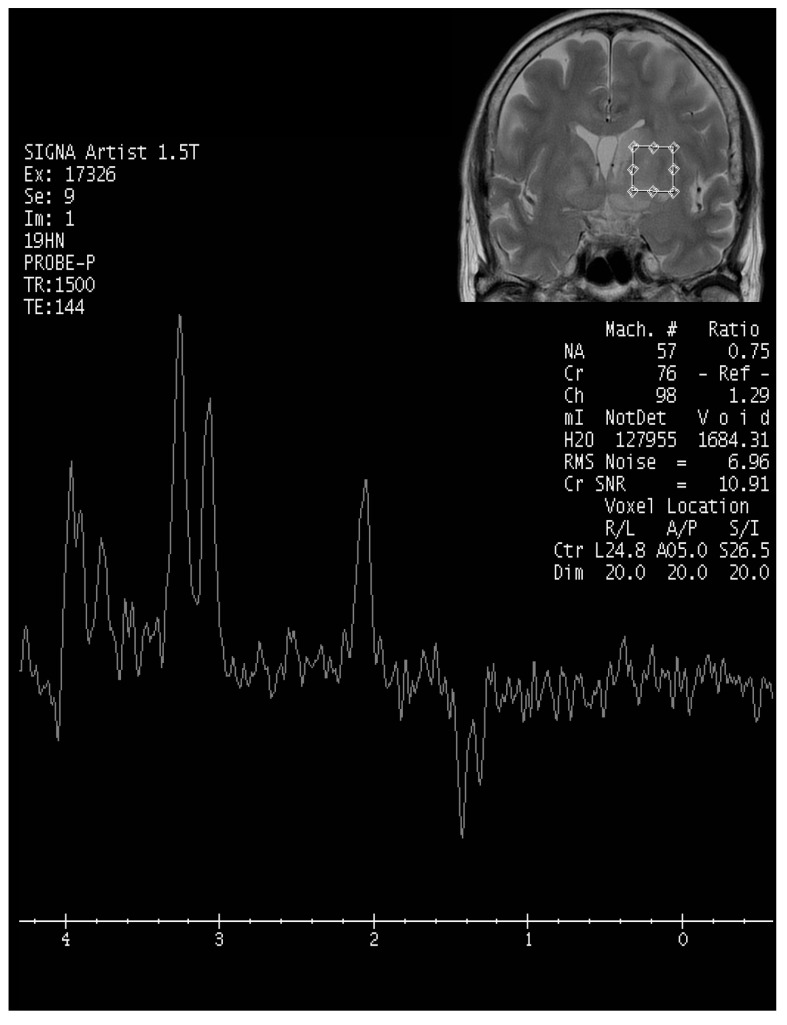
MR spectrocopy focusing on left basal ganglion lesion. Elevated choline peak and decreased N-acetylaspartate peak, with the presence of a lactate peak were found.

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
