# Peer review of "CNS Involvement of DLBCL Presenting with an Unusual Non-Enhancing Infiltrative Mass"

_diagnostics, 2023, doi:10.3390/diagnostics13223424_

Round 1

Reviewer 1 Report

Comments and Suggestions for Authors

I appreciate the opportunity to review the work entitled ‘CNS involvement of DLBCL presenting with an unusual non-enhancing infiltrative mass’. Below I describe my concerns regarding this paper:

1. While I concur with the authors that the case presented in the article is indeed quite rare, it is neither captivating nor educational. Given the patient's medical history, as well as the presence of leukopenia and neutropenia, refractory DLBCL should be one of the most immediate suspicions. The discovery of the CNS mass should therefore promptly raise the possibility of DLBCL of the CNS. The addition of the discovery of CNS mass should therefore immediately draw attention to the possibility of the DLBCL of CNS. While I agree with the possibility of this mass being a glioma, as suggested by the authors, lymphoma seems to be a much more likely option. Consequently, I fail to understand as to why such a thing should warrant a publication.

2. The unnecessary use of commercial drug names is not only unwanted, but can also hinder readability. 

3. Although I understand that this article is an interesting image, it is written in an overly concise manner, omitting many significant details.

Reviewer 2 Report

Comments and Suggestions for Authors

Dear Editor,

First, I would like to thank the authors for their work. This is an uncommon CNS manifestation of DLBCL. It deserves publication after revision.

-          Please the authors explain more about the brain biopsy. Which part of the lesion was biopsied? What type of biopsy was performed?

-          In row 61, does brain radiotherapy mean performing WBRT again (re-WBRT)?

-          In line 102, the authors have mentioned that PML does not cause a mass effect (as a difference with the reported cases), whereas in the present case, the brain lesion is infiltrative rather than a lesion with a mass effect.

Yours sincerely,

Reviewer 3 Report

Comments and Suggestions for Authors

This manuscript presents the radiological features of a patient with central nervous system involvement in diffuse large B-cell lymphoma (DLBCL). Presentation of the clinical features and radiological findings of the case is sufficient. The quality of MRI images is appropriate. The findings were discussed appropriately in light of the literature. The manuscript is suitable for publication

Round 2

Reviewer 1 Report

Comments and Suggestions for Authors

The authors have responded to all of the comments.